# Survival to Age 90 in Men: The Tromsø Study 1974–2018

**DOI:** 10.3390/ijerph16112028

**Published:** 2019-06-06

**Authors:** Tormod Brenn

**Affiliations:** Department of Community Medicine, Faculty of Health Sciences, UiT—The Arctic University of Norway, N-9019 Tromsø, Norway; tormod.brenn@uit.no; Tel.: +47-776-44-823

**Keywords:** longevity, survival, risk factors, smoking, physical inactivity

## Abstract

The 738 oldest men who participated in the first survey of the population-based Tromsø Study (Tromsø 1) in Norway in 1974 have now had the chance to reach the age of 90 years. The men were also invited to subsequent surveys (Tromsø 2–7, 1979–2016) and have been followed up for all-cause deaths. This study sought to investigate what could be learned from how these men have fared. The men were born in 1925–1928 and similar health-related data from questionnaires, physical examination, and blood samples are available for all surveys. Survival curves over various variable strata were applied to evaluate the impact of individual risk factors and combinations of risk factors on all-cause deaths. At the end of 2018, 118 (16.0%) of the men had reached 90 years of age. Smoking in 1974 was the strongest single risk factor associated with survival, with observed percentages of men reaching 90 years being 26.3, 25.7, and 10.8 for never, former, and current smokers, respectively. Significant effects on survival were also found for physical inactivity, low income, being unmarried, high blood pressure, and high cholesterol. For men with 0–4 of these risk factors, the percentages reaching 90 years were 33.3, 24.9, 12.4, 14.4, and 1.5, respectively. Quitting smoking and increasing physical activity before 55 years of age improved survival significantly. Men should refrain from smoking and increase their physical activity, especially those with low income, those who are unmarried, and those with high blood pressure and high cholesterol.

## 1. Introduction

An overwhelming number of scientific papers have been devoted to investigating the risk factors for certain diseases and death. A variety of risk factors have been identified, many of them modifiable, i.e., from which one could possibly refrain or at least partly improve or control [1]. This knowledge and resultant preventive efforts, together with better living conditions and advances in medicine, have resulted in a remarkable increase in life expectancy and in the number of people reaching older ages [2,3]. This has turned researchers’ attention to changes in the risk factors for disease and death from mid-life into late-life, as well as strategies for a long, healthy life and how to reach the ages of 85 or 90 years [4,5,6,7,8].

In Norway, as in other developed countries, the rise of coronary heart disease during the 1960s was a major health problem [9]. For men 45–49 years of age, mortality from coronary heart disease in Norway increased rapidly, from 581 per 1 million per year in 1951–1955 to 1451 per 1 million per year in 1971–1972 [10]. In response to this, studies were initiated to identify the causes of this disease, with the goal of reducing its incidence. In Tromsø, Northern Norway, male residents aged 20–49 years were invited to the Tromsø Study, a first study on coronary heart disease, in 1974. Although the first survey in this study (Tromsø 1) focused mainly on cardiovascular risk factors and diseases, a comprehensive range of general health-related information was collected from questionnaires, physical examination, and blood samples. Broader health perspectives were gradually introduced into later surveys (Tromsø 2–7) [11].

The oldest men participating in Tromsø 1 were also invited to Tromsø 2–7 and have now had the chance to reach the age of 90 years. The data collected in Tromsø 1–7, together with follow-up on all-cause deaths, provide an opportunity to investigate factors associated with longevity in men. The aim of this study was to identify risk factors, individually and in combination, and their impact on reaching up to 90 years of age. A further aim was to investigate the possible effect of changes in risk factors from middle to older age.

## 2. Materials and Methods

### 2.1. Study Population

In 1974, a total of 6595 men aged 20–49 years attended Tromsø 1. Among those were 738 men born in 1925–1928 (attendance rate 83%). Moving away from Tromsø to other places within Norway was recorded at the date of moving. Men who were still alive and living in the municipality were invited to Tromsø 2–7 between 1979 and 2016. Detailed descriptions are found elsewhere for Tromsø 1 [10], Tromsø 1–6 [11], and Tromsø 7 [12]. Approval was granted by the Data Inspectorate and by the Regional Committee and Health Research Ethics, North Norway (#REK Nord ref. 2014/940).

### 2.2. Measurements

In Tromsø 1, self-reported information was collected via questionnaire on smoking status, physical activity, income, marital status, diabetes, medication use, and unemployment. Objective measures of height and weight, blood pressure (measured by a sphygmomanometer), and blood lipids (e.g., serum total cholesterol) from a non-fasting venous blood sample were also recorded. Similar information was collected in Tromsø 2–7, with a slight reformulation in the questionnaires in some surveys. Moreover, an automatic device was used to measure blood pressure (Dinamap) and laboratory updates for cholesterol were implemented in Tromsø 2–7. Income was not included after Tromsø 1.

Smoking status was categorized as never, former, and current. Physical activity during leisure time was reported on a 4-level ordinal scale (1: inactive, i.e., reading, watching TV, or other sedentary activity; 2: walking, cycling, or other forms of exercise at least 4 hours a week, including walking or cycling to place of work, Sunday walk/stroll, etc.; 3 and 4: more rigorous activities) and then categorized as inactive (level 1) and active (levels 2–4). Low income was defined according to the 10th decile. Marital status was categorized as single, married, widow/widower, and separated/divorced. High blood pressure was defined as systolic blood pressure ≥ 140 mmHg, diastolic blood pressure ≥ 90 mmHg [13], or use of blood pressure-lowering medication. Those with serum total cholesterol > 7 mmol/L were defined as having high total cholesterol.

### 2.3. Follow Up

Follow-up for death and emigration was performed through linkage to the Population Register of Norway using the national 11-digit unique personal identification number. Date of birth, Tromsø Study attendance, and all-cause deaths were recorded to the exact day. End of follow-up was defined as date of death, date of emigration from Norway, or at the day the participant reached 90 years of age, whichever came first. During follow-up, one man emigrated in 1975 (attended Tromsø 1) and another in 2002 (attended Tromsø 1–4). These two men are included in some analyses before they were censored at their date of emigration.

### 2.4. Statistical Methods

In addition to percentages and means, a survival analysis was performed with attained age as the time variable. As the men attended Tromsø 1 in 1974, deaths were impossible before that. Kaplan-Meier plots were used to display survival curves according to various variable strata, and testing of group differences in survival was done with the logrank test, providing Chi-square statistics ($\chi_{\mathrm{df}}^{2}$, df = degrees of freedom) with *p*-values.

Current smoking, physical inactivity, low income, being unmarried, high blood pressure, and high total cholesterol all reached the 5% level for statistical significance on survival. These risk factors were considered individually and in combination to assess their joint effect. This was done by counting how many any of the 6 risk factors were present for each man. The number of missing values in Tromsø 1 was 5 for smoking status and 0–2 for the other variables. To investigate the possible benefit of changing lifestyle after Tromsø 1, survival was analyzed according to changes in smoking status and physical activity from Tromsø 1 to Tromsø 2 and from Tromsø 2 to Tromsø 3. Follow-up was considered for men who attended both surveys. For data management, analyses, and figures, we used the statistical software SAS 9.4 (SAS Institute, Cary, NC, USA) and IBM SPSS Statistics 25 (IBM Corp., Armonk, NY, USA).

## 3. Results

### 3.1. Tromsø 1–7

The youngest attendee in Tromsø 1 was aged 45.4 years and the oldest in Tromsø 7 was 91.3 years. Note that although follow-up ended when the participant reached the age of 90 years, there were still some attendees in Tromsø 7 who were beyond that age. Among the 738 men who attended Tromsø 1 in 1974, 40 attended Tromsø 7 in 2015–2016 (Table 1).

By the end of follow-up, all 736 men who attended Tromsø 1 and had not emigrated had had the chance to reach 90 years of age; the number who actually did so was 118 (16.0%). Among those 118 men, a substantial number also participated in subsequent Tromsø surveys (for example, 104 in Tromsø 3 and 103 in Tromsø 4). After that, the men were over 70 years of age and attendance rates dropped in Tromsø 5 and beyond. Although in theory all 118 men who reached 90 could have attended all subsequent surveys, the number of attendees among those not reaching 90 dropped from survey to survey, partly due to deaths. In Tromsø 1, 43.2% of the 118 men who still were alive at 90 were current smokers, compared with 68.5% among those who had died before that age. In Tromsø 2–7, the proportion of current smokers declined in both groups, but the relative proportion remained, for example, in Tromsø 3, with 28.9% of those alive at 90 being current smokers versus 53.0% of those who died before reaching that age. There were considerable differences in physical inactivity and being unmarried among men who did and did not live to 90 years of age (Table 2).

The general pattern of risk factors persisted from Tromsø 1 to Tromsø 7, with less adverse mean values among those who were alive at 90 than among those who had died. One exception was total cholesterol, which was more beneficial among those who died from Tromsø 4 onwards (Table 3).

### 3.2. Individual Risk Factors

The youngest man who died reached the age of 47.1 years. Among the lifestyle factors recorded in Tromsø 1, smoking status was by far the single strongest factor associated with survival. The median ages at death (50% survival) were 83.1, 82.7, and 75.2 years for never, former, and current smokers, respectively, and 26.3%, 25.7%, and 10.8% of men in these categories, respectively, reached 90 years of age. The pattern for the other risk factors was rather similar, with survival curves starting to diverge among men in their mid 50s, with 50% survival approximately 5 years apart. A large number of men had high total cholesterol, and a rather similar percentage of those with (15.5%) and without (17.3%) this risk factor reached 90 years of age (Figure 1).

### 3.3. Risk Factors in Combination

There were nine men with 5 risk factors and one man with all 6, and due to the small numbers, they were excluded from the analysis presented herein. The joint effect on survival clearly increased with increasing number of risk factors, and among those with 0 factors, i.e., who were “very healthy” in Tromsø 1, 33.3% reached the age of 90 compared with 12.4% among those with 3 factors. There were slightly fewer men who reached 90 among those with 2 than those with 3 risk factors (12.4% versus 14.4%), but reaching younger ages was less likely for those with 3 factors. The observed median ages at death were 85.0, 83.8, 78.1, 73.9 and 64.9 years for men with 0 to 4 risk factors, respectively. With 0 versus 4 risk factors, the median survival time was 20.1 years longer (Figure 2).

### 3.4. Changes in Risk Factors Later in Life

There were 474 men who reported to be current smokers in Tromsø 1; 414 of these men also attended Tromsø 2 and 76 reported they had quit smoking. Thus, the men who quit had done so in their late 40s or early 50s. These men had a more favorable survival curve from the age of approximately 60 years compared with the men who continued to smoke. Interestingly, those who quit smoking in their late 50s between Tromsø 2 and Tromsø 3, displayed no such survival benefit. A similar pattern also emerged for physical inactivity (Figure 3).

## 4. Discussion

### 4.1. Main Finding

The main finding of this study is the huge reduction in observed life length for men with two or more of the risk factors identified in this study, i.e., current smoking, physical inactivity, low income, being unmarried, high blood pressure, or high total cholesterol. Reported in adult mid-life, these risk factors started to take their toll as early as approximately 55 years of age, and over time, more and more lives were lost. The large effect of lifestyle characteristics at adult middle-age was further underlined by the decreasing survival time and smaller number of men reaching 90 years of age that was observed with increasing number of risk factors. Although each of these risk factors alone was associated with premature death, their massive joint effect emphasizes the benefits of eradicating as many of them as possible.

This study had data to investigate a unique combination of information from questionnaires, measurements, and blood samples. The data on individual birth and death to the exact day allowed us to present figures displaying observed survival at any age from 50 to 90 years. Our findings of considerably reduced survival related to combinations of the investigated risk factors are consistent with other reports. All-cause mortality for the combination of all risk factors versus no risk factors has been reported to range from three- to four-fold [14,15,16]. Depending on the age of the participants, the percentage of individuals reaching 85 years of age with all versus no risk factors were 2% and 37% [4], 22% and 69% [5], and 4% and 54% [6]. Estimates or projections of gained life expectancy by adopting a low-risk lifestyle have ranged from 7.4 to 17.9 years [15,16,17,18,19].

### 4.2. Combinations of Two Factors

High blood pressure and high total cholesterol have often been found to coexist and more than additively increase the risk of cardiovascular disease [20]. In our data, the correlation coefficient between systolic pressure and total cholesterol was 0.13 (not shown in results). The logrank test for the 167 men with both high blood pressure and high cholesterol had a Chi-square value of 5.18. As this was only marginally larger than the value for high cholesterol alone (5.09), in this respect, no large additional combined effect appeared. Current smoking led to an increased risk for some diseases, and this effect was strengthened by hypercholesterolemia [21]. One study reported the 24.6% of deaths from coronary heart disease were attributable to the coexistence of smoking and hypertension [22]. We found that smoking combined with high blood pressure or high cholesterol both had a lower Chi-square value than smoking alone, indicating no additional effect on survival.

### 4.3. Individual Risk Factors

This study highlights the role of smoking as the single leading modifiable cause of premature death. In Tromsø 1 (when men were 45.4–49.9 years of age), as many as 64.3% of the 738 men in our study sample were current smokers. The benefit of quitting smoking was displayed with increasing age, as the survival curve of former smokers gradually approached that of never smokers. The results herein point to a more positive gained effect the earlier smoking is ended. Others have concluded that smoking cessation is beneficial at any age [23] and should be emphasized regardless of age [24].

Physical inactivity during leisure time was reported by 27.8% of our study participants, and the survival curves for these men started to depart from that of active men as early as approximately 50 years of age. The substantial effect of being at least moderately physically active was thus confirmed by this study. Our findings correspond with others who have reported an increased probability of longer survival with regular physical activity [7,25,26]. Furthermore, moderate-intensity physical activity has been found to increase life expectancy [27,28].

Men with low income started to die early, and this may be due to the fact that 5.6% had no income at all, indicating that some health problem was present in Tromsø 1. As income was included in Tromsø 1 only, analyses and interpretation of the impact of this risk factor or its progress on survival were restricted. The number of men in Tromsø 1 who were receiving unemployment benefits was modest (*n* = 16). Education was included in Tromsø 2 in a second questionnaire returned by approximately 90% of the attendees, and the percentages with minimum education (7 years or less) were 39.6 and 46.8 for those alive and dead at the age of 90 years, respectively. Our results thus support those of others, showing that mortality is associated with income, education, and socioeconomic status [4,5].

Interestingly, the variety of risk factors documented in this study also included marital status. Unfortunately, due to the low numbers in Tromsø 1 of widowers (*n* = 6) and separated/divorced (*n* = 22), these groups of men could not be analyzed for survival separately. Losing a spouse has been documented to increase mortality [29], and a consistent survival advantage for married over unmarried individuals has also been found [30].

Whereas only 4.5% of our study sample used antihypertensive medication in Tromsø 1, 38.5% did so in Tromsø 5 (when men were 72.5–76.9 years of age). Those who died before 90 on average had higher blood pressure than those who survived to that age. The increased mortality associated with high blood pressure found herein agrees with other investigations of older men [4,5,6,7,8].

The reason why the percentages of men with high total cholesterol who did and did not reach 90 years of age were so similar remains unclear. However, those who died before reaching 90 compared with those who survived to 90 had higher cholesterol means in Tromsø 1–3, but in Tromsø 4–7, this trend was reversed. In Tromsø 5, lipid-lowering medication was used by 28.6% of the men with high and 12.2% of the men with lower total cholesterol. Currently, there is debate on the challenge of how to interpret lipid-lowering medications and all-cause mortality [31,32].

### 4.4. Factors that did not Reach Statistical Significance

Several other potential risk factors were tested for survival but were not statistically significant. This was the case for body mass index, for which the difference between those alive and those dead at 90 years of age was no larger than 0.2 kg/m^2^ in Tromsø 1–6. Dichotomizing body mass index values into overweight or obesity also had no significant effect on survival. History of heart disease among a first-degree relative was also not significant. Other variables were reported by few men in Tromsø 1, such as diabetes (*n* = 4) and blood pressure treatment (*n* = 33).

### 4.5. Strengths and Weaknesses

A strength of this study is that it is population-based with a high attendance rate and used standardized methods. The number of missing values was negligible, as the completion of questionnaires was controlled at the surveys. As opposed to most studies in this field, which are based on questionnaire information only, we had objective measurements of blood pressure and blood lipids. Obviously, given the duration of the study, with Tromsø 1 in 1974 to Tromsø 7 in 2015–2016, some changes in instruments and methodology were unavoidable. However, most routines have been kept identical for comparison purposes.

The results herein were based on observed data, not on estimates or predictions. Massive changes have occurred since Tromsø 1 in 1974. From 1973 to 2014, daily smoking dropped from 51% to 14% among Norwegian men [33]. During these decades, a substantial decline was also seen in blood pressure [34], total cholesterol [12], and cardiovascular deaths among Tromsø Study attendees [35]. These changes are likely to provide a somewhat different risk pattern for middle-aged men today. But, despite these time changes, our results may still serve as a guideline for how to obtain a long life.

A limitation is that data on women not were collected; nor did we have data on some factors that could have had a significant impact on survival. Such examples include mental factors [36] and diet [14], both of which have been documented to be important for survival. Questions on alcohol consumption were not included in Tromsø 1. The question was introduced in the second questionnaire in Tromsø 2, and 12.9% and 8.7% of men alive and dead, respectively, at 90 years of age reported being teetotallers. This indicates an adverse effect of alcohol in this study, which has also been documented by others [37].

## 5. Conclusions

This study has shown that, if one wants to live a long life, smoking should be avoided and regular physical activity should be adhered to. Furthermore, those having only one adverse factor should be careful, but among those with two or more adverse factors, a major change in lifestyle is needed. Finally, questionnaire and measured risk factors collected at adult mid-life have a considerable ability to predict longevity.

## Figures and Tables

**Figure 1 ijerph-16-02028-f001:**
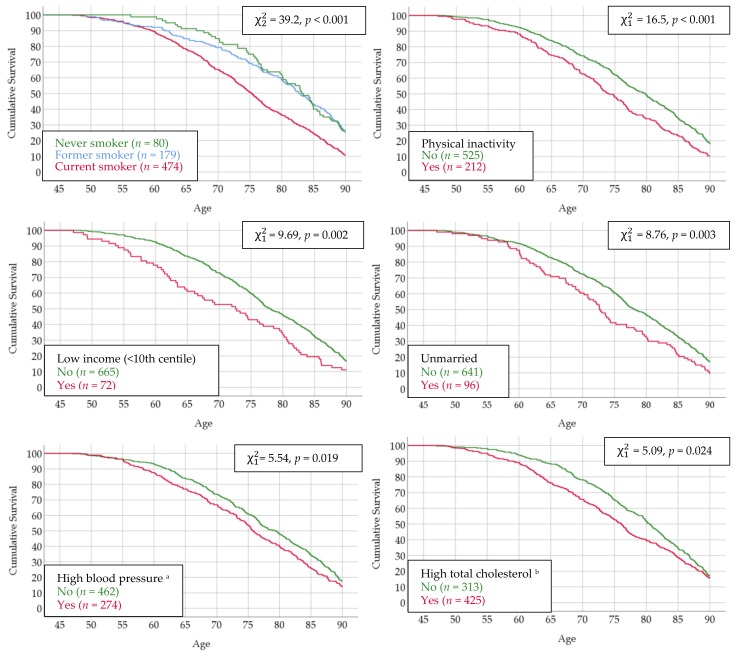
Survival curves according to risk factors in Tromsø 1. The Tromsø Study. ^a^ Systolic blood pressure ≥ 140 or diastolic pressure ≥ 90 or blood pressure medication; ^b^ Total cholesterol > 7.

**Figure 2 ijerph-16-02028-f002:**
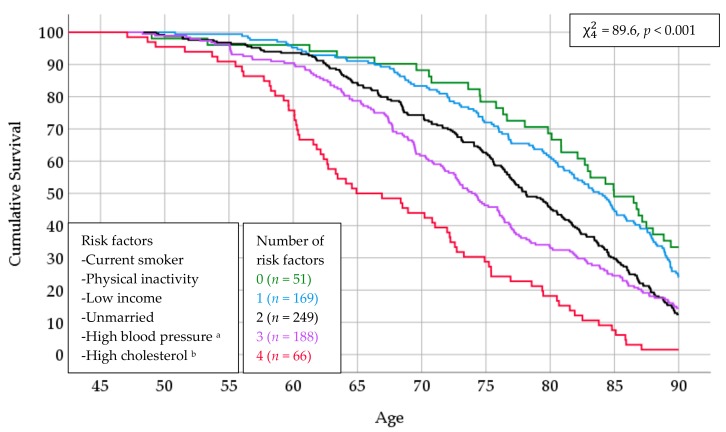
Survival curves according to number of risk factors in Tromsø 1. The Tromsø Study. ^a^ Systolic blood pressure ≥ 140 or diastolic pressure ≥ 90 or blood pressure medication; ^b^ Total cholesterol > 7.

**Figure 3 ijerph-16-02028-f003:**
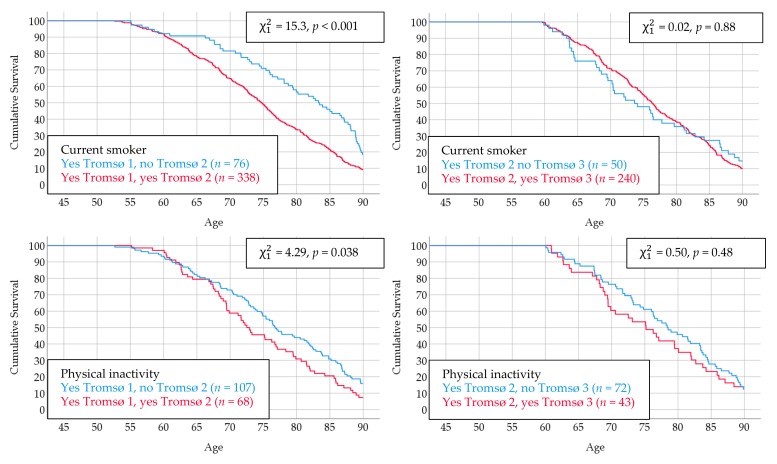
Survival curves according to changes in risk factors between Tromsø 1 and Tromsø 2 or between Tromsø 2 and Tromsø 3. The Tromsø Study.

**Table 1 ijerph-16-02028-t001:** Number of attendees, attained age range, and mean age at survey. The Tromsø Study.

Survey	Year(s) of Survey	Attended ^1^	Age Range of Attendees (Years)	Mean Age of Attendees (Years)
Tromsø 1	1974	738	45.4–49.9	47.7
Tromsø 2	1979–1980	642	50.8–54.9	53.0
Tromsø 3	1986–1987	571	57.8–62.1	60.0
Tromsø 4	1994–1995	464	65.9–70.3	68.2
Tromsø 5	2001–2002	311	72.5–76.9	74.6
Tromsø 6	2007–2008	139	79.1–83.7	81.2
Tromsø 7	2015–2016	40	86.5–91.3	88.6

^1^ Included is one man who attended Tromsø 1 and another who attended Tromsø 1–4 before they emigrated from Norway.

**Table 2 ijerph-16-02028-t002:** Prevalence of risk factors according to being alive or dead at age 90 years by survey. The Tromsø Study.

Survey	Number of Men Who Attended		Current Smoker (%)		Physical Inactivity (%)		Unmarried (%)	
	Alive	Dead	Alive	Dead	Alive	Dead	Alive	Dead
Tromsø 1	118	618	43.2	68.5	18.6	30.6	7.6	14.1
Tromsø 2	108	533	31.5	59.7	13.9	21.8	8.3	14.8
Tromsø 3	104	466	28.9	53.0	15.4	22.8	10.6	17.8
Tromsø 4	103	360	23.3	30.1	18.5 ^1^	22.2 ^1^	15.5	23.3
Tromsø 5	89	222	17.1	22.1	11.2 ^1^	23.0 ^1^	15.7	26.6
Tromsø 6	71	68	10.0	23.0	12.7	20.6	25.3	39.7
Tromsø 7	34	6	6.3	16.8	17.7	16.7	40.0	33.3

^1^ A different question was used. None or less than 1 hour per week with light activity (not sweating or out of breath).

**Table 3 ijerph-16-02028-t003:** Means of risk factors according to being alive or dead at age 90 years by survey. The Tromsø Study.

Survey	Number of Men who Attended		Total Cholesterol (mmol/L)		Systolic Blood Pressure (mmHg)		Diastolic Blood Pressure (mmHg)	
	Alive	Dead	Alive	Dead	Alive	Dead	Alive	Dead
Tromsø 1	118	618	7.31	7.38	128.1	131.1	80.3	83.3
Tromsø 2	108	533	6.66	6.85	131.3	137.5	84.3	88.7
Tromsø 3	104	466	6.47	6.62	137.4	143.5	83.1	86.6
Tromsø 4	103	360	6.52	6.48	147.5	153.7	84.6	87.3
Tromsø 5	89	222	6.01	5.87	148.4	152.6	81.2	82.9
Tromsø 6	71	68	5.06	5.03	155.5	156.3	79.5	79.2
Tromsø 7	34	6	5.04	3.72	145.5	149.2	72.0	73.2

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
