# Peer review of "Survival to Age 90 in Men: The Tromsø Study 1974–2018"

_ijerph, 2019, doi:10.3390/ijerph16112028_

Round 1
Reviewer 1 Report
The manuscript’s topic is very persuasive and warrants publication. Assessing survival curves over various variables and evaluating the impact of individual risk factors and combinations of risk factors of death with overtime with a cohort of males is very relevant to life expectancy research.
The research findings that smoking and physical inactivity have significant effects on survival, and it possesses global implications with on current priority health prevention and intervention strategies. The manuscript was well written and the research methods (design, sample size and statistical analysis) were appropriate, and more important the study can be replicated.
However, In the limitation section, the lack of marital status stratification needs to also be considered as a limitation. Marital status and mortality (married vs separated or widowed) garner various implications on the longevity of males.
Author Response
Reviewer comment:
"However, In the limitation section, the lack of marital status stratification needs to also be considered as a limitation. Marital status and mortality (married vs separated or widowed) garner various implications on the longevity of males."
I agree and have added page 7, lines 223-224:
"Unfortunately, due to the low numbers in Tromsø 1 of widowers (n=6) and separated/divorced (n=22), these groups of men could not be analyzed for survival separately."
Reviewer 2 Report
This study identified the relevant factors, individually and in combination, and their impact on reaching up to 90 years of age among male. This is an interesting observation, the statistics used are appropriate, the conclusions derived from these, and the interpretations are consistent and sound. However, the manuscript still needs a few revisions to be acceptable for the publication in International Journal of Environmental Research and Public Health as an Article.
Minor comments:
Title: The participants of this study were only men. Please make an addition so that we can understand it even if we read only the title.
Figure 3 (the bottom right): Is the note “Yes Tromsø 1, yes Tromsø 2 (n=43)” correct? I suppose “Yes Tromsø 2, yes Tromsø 3 (n=43)”.
Discussion (P7, L190-192): Where is the result shown here displayed in the results chapter?
Author Response
Minor comments:
"Title: The participants of this study were only men. Please make an addition so that we can understand it even if we read only the title."
I agree and have changed the title from "Survival to Age 90 Years: The Tromsø Study 1974–2018" to "Survival to Age 90 in Men: The Tromsø Study 1974–2018".
"Figure 3 (the bottom right): Is the note “Yes Tromsø 1, yes Tromsø 2 (n=43)” correct? I suppose “Yes Tromsø 2, yes Tromsø 3 (n=43)”."
My mistake, and the reviewer is absolutely correct. I have changed as indicated.
"Discussion (P7, L190-192): Where is the result shown here displayed in the results chapter?"
I agree, and have now added in page 7, line 192: (not shown in results).